# Modulation of Gut Microbiota for the Prevention and Treatment of COVID-19

**DOI:** 10.3390/jcm10132903

**Published:** 2021-06-29

**Authors:** Jiezhong Chen, Luis Vitetta

**Affiliations:** 1Medlab Clinical, Research Department, Sydney 2015, Australia; jiezhong_chen@medlab.co; 2Faculty of Medicine and Health, The University of Sydney, Sydney 2006, Australia

**Keywords:** COVID-19, intestinal dysbiosis, butyrate, probiotics, nutraceuticals

## Abstract

The gut microbiota is well known to exert multiple benefits on human health including protection from disease causing pathobiont microbes. It has been recognized that healthy intestinal microbiota is of great importance in the pathogenesis of COVID-19. Gut dysbiosis caused by various reasons is associated with severe COVID-19. Therefore, the modulation of gut microbiota and supplementation of commensal bacterial metabolites could reduce the severity of COVID-19. Many approaches have been studied to improve gut microbiota in COVID-19 including probiotics, bacterial metabolites, and prebiotics, as well as nutraceuticals and trace elements. So far, 19 clinical trials for testing the efficacy of probiotics and synbiotics in COVID-19 prevention and treatment are ongoing. In this narrative review, we summarize the effects of various approaches on the prevention and treatment of COVID-19 and discuss associated mechanisms.

## 1. Introduction

Severe COVID-19 is characterized by both hyperinflammation and lymphopenia [1]. As such, it could affect multiple organs, causing acute respiratory distress syndrome (ARD), multi-organ failure, and a high rate of mortality. Multiple risk factors for severe COVID-19 have been identified including advanced age, underlying chronic diseases, and use of antibiotics. Dysregulation of the gut microbiota (gut dysbiosis) is associated with the hyperinflammation caused by risk factors in severe COVID-19 disease. Gut dysbiosis can cause chronic inflammation and decreased anti-inflammatory mechanisms to facilitate the formation of the hyperinflammation observed in SARS-CoV-2 infections [1]. It has recently been postulated that intestinal dysbiosis in patients with COVID-19 may describe an enhanced pro-inflammatory tone in the gut with the cytokine storm [2]. This could be attributed to reduced production of commensal bacterial metabolites. Therefore, the modulation of gut microbiota and supplementation of commensal bacterial metabolites could reduce the hyperinflammatory response in severe COVID-19.

The modulation of the intestinal microbiota can be achieved by probiotics, prebiotics, and synbiotics (Figure 1). Supplementation with nutraceuticals and trace elements could also reduce the severity of COVID-19 by improving gut microbiota abundance and diversity. In addition, commensal bacterial metabolites could be supplied to patients with COVID-19, particularly in cases with gut dysbiosis caused by gut SARS-CoV-2 infections and the administration of antibiotics [1].

In this narrative review, we summarize preclinical studies of various approaches to improve the gut microbiota, discuss associated mechanisms, and posit possible effects on the prevention and treatment of COVID-19 infections.

## 2. Probiotics

The rationale for the use of probiotics is from the association of gut dysbiosis and the severity of COVID-19. Several studies have shown the dysbiosis in COVID-19 with decreased beneficial bacteria *Lactobacilli* and *Bifidobacteria* as well as butyrate-producing bacteria *Faecalibacterium prausnitzii* and *Eubacterium rectale* [3,4]. Alternatively, opportunistic pathogenic bacteria such as *Clostrium ramosum* and *Clostridium hathewayi* are increased in abundance [4]. Gut dysbiosis is correlated with an increased cytokine storm and disease severity. The abundance of *Faecalibacterium prausnitzii* is negatively correlated with the severity of COVID-19, whereas *Clostridium ramosum* and *Clostridium hathewayi* are positively correlated with the severity of COVID-19 [4]. Therefore, restoration of these adverse microbiome shifts should be helpful in reducing the severity of COVID-19, and as such, the intestinal dysbiosis that mimics that observed in other clinical scenarios can be relieved by supplementation with probiotics [5]. Indeed, so far, 19 clinical trials are ongoing to test the efficacy of probiotics and synbiotics on the prevention and treatment of COVID-19 (ClinicalTrials.gov, accessed on 15 May 2021) (Table 1).

In these clinical trials, the most commonly employed strains were from the *Lactobacilli* and *Bifidobacteria* genera which are decreased in COVID-19. *Bifidobacteria*, belonging to the Actinobacteria phylum, are Gram positive, anaerobic bacteria, which produce acetate, secrete exopolysaccharides, and protect hosts from increased pathobiont proliferation and assault [6]. *Bifidobacteria* are abundant in infants but decrease with aging. The decreased abundance of *Bifidobacteria* in advanced age is associated with many chronic diseases [7]. *Lactobacilli* are Gram-positive, anaerobic, rod-like bacteria which can secrete lactic acid to inhibit pathogenic bacteria [8]. *Lactobacilli* contain more than 250 member species, a large number of which are employed in probiotic formulation development. Indeed, various *Lactobacillus*-based probiotics are available which show beneficial effects on many chronic inflammatory diseases such as obesity, diabetes, and hypertension [8].

Probiotics could exert their anti-viral and anti-inflammatory effects to facilitate COVID-19 prevention and treatment. Peptides produced by some probiotics such as *Lactobacilli* and *Paenibacillus* have been demonstrated to bind to ACE2, blocking the binding of SARS-CoV-2 to targeted cells [9,10]. Probiotics have also been shown to elicit anti-viral immunity to facilitate the elimination of viruses. In a mouse influenza model, *Lactobacillus rhamnosus* GG stimulated TLR4 signal to increase anti-viral response [11]. Another study showed that *Lactobacillus gasseri* increased IFN type I and II secretion, reducing respiratory syncytial viral infections in a mouse model [12]. In thirty heathy elderly patients, administration of *Bifidobacterium lactis* increased phagocytic and lytic activities of monocytes and killer cells [13]. Sophisticatedly, probiotics have been engineered to carry anti-microbial peptides such as cathelicidin-encoding *Lactococcus lactis* [14]. Therefore, the use of probiotics in COVID-19 could facilitate blocking the virus from binding to the host cells, reduce viral replication, and increase host immunity against the virus.

The anti-inflammatory effects of probiotics could reduce the hyperinflammation in COVID-19. Probiotics have been demonstrated to reduce inflammation in in vitro and in vivo studies in various inflammation-inducing models [15]. In a high-fat diet-induced obese mouse model, *Lactobacillus fermentum* CECT5716 reduced inflammation markers and improved epithelial cell functionality [16]. In an intestinal inflammation mouse model, *Lactobacilli* reduced the levels of TNF-α and increased production of IL-10 [17]. *Lactobacilli* also increased gut barrier. *Bifidobacterium bifidum* expressing IL-10 reduced IL-6 production in RAW 264.7 stimulated by LPS and IL-8 production in HT-29 cells stimulated by LPS or TNF-α [18].

The improvement of gut microbiota by probiotics could increase the efficacy of vaccines against SARS-CoV-2 infections. A healthy gut microbiota is necessary for vaccine efficacy [19,20]. Studies have demonstrated that vaccines are unable to elicit robust immune responses in germ-free mice or antibiotics-administered mice [21]. A clinical trial is currently investigating a yeast-based probiotic called ABBC1 to enhance a vaccine against COVID-19 [22]. ABBC1 includes β1,3/β1,6-glucans and inactivated *Saccharomyces cerevisae* as well as trace elements selenium and zinc. It is expected that supplementation could change gut microbiota to increase COVID-19 vaccine efficacy. The trace elements included in this trial can also modulate gut microbiota (discussed in forthcoming sections). Currently, many vaccines against SARS-CoV-2 have been developed for the prevention of COVID-19. Further enhancement of their efficacies by modulation of the gut microbiota warrants additional focused studies. Furthermore, commensal bacteria such as *Enterococcus faecium* have been reported to markedly reduce the replication of coronavirus transmissible gastroenteritis virus in a swine testicle cell culture system [23].

## 3. Commensal Bacterial Metabolites

Commensal intestinal bacteria can produce various metabolites to mediate health benefits. The metabolites include short-chain fatty acids (SCFAs), bile acid derivatives, and amino acids [24]. SCFAs, particularly butyrate has been extensively studied as a mediator of commensal bacteria. In COVID-19, these beneficial metabolites could be in deficit due to dysbiosis. Several reports have proposed the use of butyrate for reducing the severity of COVID-19 infections [1,25,26]. Explicitly, butyrate could be administered in those cases with severe intestinal SARS-CoV-2 infections and/or the extensive use of antibiotics or other therapeutic agents that affect gut microbiota, such as proton pump inhibitors, antidepressants, laxatives, and statins [27,28], which makes it difficult for probiotics to help restore the abundance and diversity of the gut microbiota.

Butyrate could increase immune responses to protect hosts from SARS-CoV-2 infections. Butyrate has direct anti-viral effects through increased secretion of mucins and anti-microbial peptide defensins [1]. A study has shown that administration of SCFAs to animals can increase expression of IFN-γ and granzyme B in cytotoxic lymphocytes through inhibition of histone deacetylases (HDACs) [29]. All three SCFAs have such an effect but butyrate is strongest, propionate second, and acetate weakest. Another study revealed that HDAC inhibitors butyrate and panobinostat could suppress ACE2 [30]. In a gut organoid model, butyrate treatment decreased the expression of both *Ace2* and *Tmprss2* genes [31].

Butyrate can reduce the severity of COVID-19 through its well-known anti-inflammatory effect. It reduces gut inflammation by activating regulatory T cells and reduces systemic inflammation through increasing the gut barrier to prevent the translocation of endotoxins and bacteria to extra-intestinal organs [28]. Butyrate also inhibits multiple pro-inflammatory signaling pathways to reduce cytokine storms [32].

## 4. Prebiotics

Prebiotics are substances that can stimulate the growth of gut commensal bacteria and produce microbial metabolites [33]. There are a wide range of prebiotics, which have been extensively studied for health benefits through gut microbiota modulation and stimulation for the production of commensal bacterial metabolites. Common prebiotics include dietary fibers, inulin, fructan, β-glucans, and arabinoxylan [33]. Unfermented dietary fibers such as cellulose cannot be considered prebiotics. Although these prebiotics have not been tested in patients with COVID-19, they are potentially effective for reducing the severity of COVID-19 through characteristic properties that modulate the intestinal microbiota. Several clinical trials in COVID-19 have included prebiotics. The beneficial effects of common prebiotics are introduced below.

### 4.1. Inulin, FOS (Fructo-Oligosaccharides), and GOS (Galacto-Oligosaccharides)

Inulin is a natural polymer of fructose linked by β (2,1) glycosidic bond, which exists in many plants such as Jerusalem artichoke, onion, garlic, wheat, and asparagus as energy storage [34]. Inulin can have as many as 200 fructose residues while it refers to fructo-oligosaccharide if the fructose residue number is less than 10. An intervention study in healthy subjects showed that administration of inulin (20 g daily for 6 weeks) increased *Bifidobacterium* and *Lachnispiraceae* with increased butyrate, isobutyrate, and isovalerate productions in the gut [35]. Inulin was also reported to have anti-influenza effects through the activation of CD8^+^ T cells [36]. Inulin-fed mice had increased systemic Ly6c^−^ patrolling monocytes, which in turn increased activated macrophages with decreased production of CXCL1 in the respiratory tract. Decreased CXCL1 led to decreased neutrophil recruitment to the respiratory tract and thus limited inflammation. Dietary fiber also increased butyrate production, which boosted effector CD8^+^ T cells to increase virus clearance. In a feline experiment, feeding of 1-kestose (the simplest fructo-oligosaccharides with two fructose residues) increased the abundance of butyrate-producing bacteria as well as *Bifidobacterium* [37]. In a mouse obese model established by subcutaneous injection of monosodium glutamate, administration of 5% FOS reduced chronic inflammation, steatohepatitis, and visceral adiposity with increased fecal concentrations of SCFAs [38]. FOS increased not only cecal butyrate but also portal and aortic butyrate concentrations [39]. A clinical trial showed that FOS increased probiotic bacteria *Bifidobacterium* and *Lactobacillus* as well as butyrate-producing bacteria *Faecalibacterium*, *Ruminococcus*, and *Oscillospira* [40]. Therefore, the use of inulin and FOS in COVID-19 could restore gut microbiota abundance and diversity with increased butyrate production.

GOS is a polymer of galactose linked by β (1,4) glycosidic bond, which cannot be digested by humans but can be fermented by commensal bacteria in the intestines [34]. It is the substrate of commensal bacteria *Lactobacilli* and *Bifidobacterium*. Thus, GOS can increase the abundance of *Lactobacilli* and *Bifidobacteria* [41,42,43,44]. Not surprisingly, GOS is able to reduce infections by pathobionts. Searle et al. (2009) showed that GOS reduced *Salmonella* infections in mice [45]. Zou et al. (2020) demonstrated that GOS reduced infections of pathogenic *E. Coli* through the modification of commensal bacteria and the gut barrier [46]. GOS also decreased the inflammatory markers IL-6, TNF-α, and IL-1β. The anti-inflammatory effect may be related to butyrate production as an in vitro study showed GOS increased butyrate production [47]. In a rotavirus infection suckling rat model, FOS and GOS reduced infectivity of the virus and symptoms in rats [48].

FOS and GOS could plausibly increase the efficacy of vaccines against COVID-19. A study showed that maternal FOS supplementation increased offspring responses to an oral vaccine with increased IFN-γ and IgA as well as butyrate levels [49]. In a mouse model of influenza vaccination, a combination of 2-fucosyllactose, FOS, and GOS increased immune responses, including increased vaccine-specific delayed type hypersensitivity, increased IG1/IG2a in serum and activated B cells, regulatory T cells, and Th1 cells in mesenteric lymph nodes [50]. Supplementation of FOS also reduced the allergy of mice to bovine milk whey protein through the activation of regulatory T cells [51], indicating that FOS may be able to reduce the side effects of vaccines against COVID-19.

### 4.2. β-Glucans

β-glucans are glucose polymers linked by β(1,3)-glycosidic bonds as backbones branched with β(1,6) bonds, which are mainly derived from the cell walls of yeast, mushroom, and oats [52]. β-glucans are well studied to have immune modulation effects to promote both innate and adaptive immune responses [52]. Studies have shown that β-glucans are also designated as prebiotics, which promote beneficial bacterial growth and increase production of SCFAs. Dong et al. (2020) showed that oat β-glucan promoted the growth of *Lactobacillus*, *Bifidobacterium*, and butyrate-producing bacteria *Blautia* and *Dialister* with increased production of butyrate in an in vitro human fecal microbiota fermentation system [53]. In weaned pigs, feeding of oat β-glucan promoted the growth of *Lactobacilli* and *Bifidobacteria* with increased production of butyrate [54]. In rats, feeding of oat β-glucan together with resistant starch increased acetate, propionate, and butyrate levels [55]. A clinical trial showed that 3% barley β-glucans in diet increased beneficial bacteria *Clostridium orbiscindens*, *Clostridium sp*., *Roseburia* hominis, and *Ruminococcus sp.* with markedly increased SCFAs including 2-methyl-propanoic, acetic, butyric, and propionic acids. β-glucan hydroxylates have a higher ability to increase the production of butyrate [56], which warrants further studies and clinical trials.

### 4.3. Resistant Starches

Resistant starches cannot be digested by human enzymes amylases and amyloglucosidases but can be fermented by gut microbiota in the intestines [57]. There are five types of resistant starches, namely, type 1 (RS1)—in food matrix or seed; type 2 (RS2)—in compact granules; type 3 (RS3)—retrogradation by cooking/cooling; type 4 (RS4)—chemical modification by food manufacture; and type 5 (RS5)—forming complex with lipid [53]. In a mouse model fed with a high-fat diet, addition of RS2 in the diet increased the colonic butyric acid concentrations by 2.6-fold with decreased inflammation, weight gain, and hepatic steatosis [58]. Several clinical trials showed that RS2 increased abundance of *Bifidobacteria* and increased production of SCFAs [59]. RS3 produced large amounts of butyric acid after fermentation in vitro [60]. A recent study in an in vitro fermentation system showed that RS5 was superior in the production of butyrate while RS3 produced more lactic acid [61]. These studies indicate that resistant starches could modulate gut microbiota and improve butyrate levels in COVID-19.

## 5. Synbiotics

Synbiotics have greater advantages in modulating the gut microbiota when compared with probiotics and prebiotics. The gut fermentation production of commensal bacteria requires both commensal bacteria and substrates for fermentation. Synbiotics provide both needs to facilitate the production of commensal bacterial metabolites. In a mouse colitis model, Son et al. (2019) showed that a synbiotic composed of *Lactobaccillus* GG and tagatose was superior to *Lactobaccillus GG* and tagatose alone in inhibiting inflammatory responses [62]. Fuhren et al. (2021) showed that inulin increased the intestinal persistence of inulin consuming bacterium *L. plantarum Lp900* contained in the synbiotic [63]. Studies also showed that the synbiotic *Bacillus coagulans* MTCC5856 and sugarcane fiber produced more profound effects than *B. coagulans* and sugarcane fiber alone in inhibiting inflammation and producing SCFAs in a colitis model [64,65].

In a clinical trial, a synbiotic called Omni-Biotic^®^ 10 AAD was used for COVID-19 treatment. It contained two *Bifidobacterial* strains, one *Enterococcus* strain, and seven *Lactobacillus* strains [66]. The prebiotics used were maize starch, inulin, and FOS. The synbiotic also contained trace elements magnesium sulphate, manganese sulphate, and potassium chloride. The effectiveness of Omni-Biotic^®®^ 10 AAD was demonstrated in a previous clinical trial for the treatment of sepsis [67]. It improved gut microbiota diversity and the gut barrier, and reduced inflammation. A similar synbiotic (Omni-Biotic^®^ stress repair) was shown to have anti-inflammatory effect in inflammatory bowel disease [68] with increased butyrate production, reduced CD4^+^ T cells, and increased gut barrier. The anti-inflammatory effect of the synbiotic could be mediated by regulatory T cells, which are activated by butyrate [69].

## 6. Nutraceuticals

Nutrition status has been associated with severity of COVID-19 infections. Moreover, in the elderly hospitalized with COVID-19, the Geriatric Nutritional Risk Index was shown to be a significant predictor of survival [70]. Deficiencies in nutraceuticals can result in severe COVID-19. Therefore, supplementation with nutraceuticals that have immune-boosting capabilities (e.g., lactoferrin, selenium, quercetin) [71] could be adjunct therapies to relieve COVID-19 infection severity and reduce the rate of mortality. Certainly, additional studies are required to fully understand the putative effects of nutraceuticals against COVID-19 severity of infections [72].

### 6.1. Vitamin C

Vitamin C was found to have anti-viral effects during the early 1970s, decreasing the severity of the common cold [73,74]. In a mouse model, supplementation of vitamin C increased anti-viral effects through increased production of IFNα and IFNβ [75]. These findings lead to suggestions of vitamin C for the treatment of COVID-19. Indeed, a study showed that vitamin C can reduce mortality rate and oxygen support in COVID-19 in a high dose [76]. So far, vitamin C has been included in more than 50 clinical trials for the treatment of COVD-19 infections (ClinicalTrials.gov, accessed on 15 May 2021).

Vitamin C supplementation can improve gut microbiota and butyrate production [77]. In an animal model, vitamin C improved hypertension, which is a risk factor for severe COVID-19, through improvements in the gut microbiota, gut integrity, and reducing inflammatory sequelae [78,79].

### 6.2. Vitamin D

Gavioli et al. (2021) [80] and others have reported that deficiency of vitamin D is associated with susceptibility to SARS-CoV-2 infections, severity of COVID-19, and mortality rate [1,79,81]. Katz et al. (2021) examined the strength of such an association and found that vitamin D deficiency increased the likelihood of positive COVID-19 by 4.6 times [81]. A recent meta-analysis study showed that deficiency of vitamin D was correlated with the severity of COVID-19; patients with poor prognosis had much lower levels of vitamin D [82]. In a retrospective study, Infante et al. (2021) reported that total serum 25-hydroxy-vitamin D levels were much higher in survivors than non-survivors (12 ng/mL vs. 8 ng/mL). Moreover, a multivariate analysis showed an inverse correlation between 25-hydroxy-vitamin D and risk of mortality [82]. Gavioli et al. (2021) also showed that 25-hydroxy-vitamin D levels less than 10 ng/mL had high hospitalization rates (98%) and mortality rates (49%) [80]. The role of vitamin D in COVID-19 could be related with its anti-inflammatory effect. Vitamin D is metabolized into 25-hydroxy vitamin D and active form 1,25-dihydroxy-vitamin D. The binding of 1,25 dihydroxy vitamin D to its intracellular receptor VDR has been postulated to regulate immune responses. As VDR exists in many cells, vitamin D can regulate many physiological processes to exert anti-inflammatory effects (Figure 2) [83,84].

There are bidirectional interactions between vitamin D and the gut microbiota, in terms of significantly influencing the abundance and diversity of the gut microbiota. In a recent study, Charoenngam et al. (2020) showed that administration of 25(OH)D increased the abundance of *Bacteroides* and *Parabacteroides* in a dose-dependent manner [85]. Alternatively, the gut microbiota can increase vitamin D levels through inducing increases in the levels of VDR receptors. In a zebrafish model, probiotic *Lactobacillus casae* BL23 increased the expression of VDR [86]. Lu et al. (2020) showed that conditional medium from five probiotic lactic acid bacteria increased VDR expression in HCT116 cells [87]. Probiotics *Lactobacillus rhamnosus* GG and *Lactobacillus plantarum* increased VDR proteins in mouse and human epithelial cells as well as expression of the target gene cathelicidin [88]. Knockout of VDR resulted in the loss of the protective effect of these probiotics on *Salmonella*-induced colitis [89].

### 6.3. Lactoferrin

Lactoferrin is an iron chelating milk bioactive peptide. Due to its beneficial effects on the development of the gut as well as disease prevention and therapy through anti-inflammatory and anti-viral effects, lactoferrin has been suggested as a therapy for COVID-19 [90,91]. An in vitro study in Caco-2 cells showed that lactoferrin inhibited SARS-CoV-2 replication through promoting anti-viral gene expression and reducing the expression of genes necessary for viral replication including RNA-dependent RNA polymerase (*RdRp*) and E gene (*CoVE*) [92]. Another in vitro study showed that bovine lactoferrin bound to heparin sulfate proteoglycan, an attachment factor for the binding of SARS-CoV-2 to ACE2, reducing the viral entry into cells [93].

Lactoferrin can modulate the intestinal microbiota by stimulating the growth of beneficial bacteria and inhibiting pathogenic bacteria. Lactoferrin and its fragment Lactoferricin B were tested for their effects on anaerobic opportunistic pathogens *Bacteroides fragilis* and *Bacteroides thetaiotaomicron* [94]. Lactoferrin but not Lactoferricin B was able to inhibit in vitro biofilm formation and binding to the laminins of these two bacteria. Other studies also reported the properties of lactoferrin and the inhibition of biofilm formation by *Bacteroides fragilis* [95,96]. In suckling piglets, feeding of lactoferrin increased abundance of *Roseburia* and decreased *Escherichia-Shigella* [96]. The colon butyrate concentrations were also increased with increased gut barrier integrity and increased secretion of mucin. In addition, the concentration of anti-inflammatory cytokine IL-10 in the colonic mucosa was increased while the concentrations of pro-inflammatory cytokines IL-1α and IL-1β were decreased. This study suggests that lactoferrin can increase gut barrier functional integrity and decrease inflammatory triggers through the modulation of gut bacteria and the secretion of bacterial metabolites such as butyrate. Therefore, supplementation of lactoferrin could reduce infectivity of SARS-CoV-2 and improve the overall gut microbiota, reducing the severity of COVID-19. At present, several clinical trials are ongoing to examine the efficacy of lactoferrin in the prevention and treatment of COVID-19 (ClinicalTrials.gov, accessed on 15 May 2021).

### 6.4. Omega-3 Fatty Acids

Omega-3 polyunsaturated fatty acids (PUFA) are well-known micronutrients benefiting human health with anti-inflammatory and anti-thrombotic effects [97,98]. However, omega-6 PUFA is pro-inflammatory and thus, the ratio of omega-3 to omega-6 has been used to indicate an adverse inflammatory state. In severe COVID-19, it was found that metabolites of omega-6, such as eicosanoids including prostanoids and leukotrienes, are increased [99]. What Hammock and colleagues regarded as an eicosanoid storm in COVID-19 promoted hyperinflammation [100]. A recent study showed that omega-3 PUFA could bind to protein spikes in its close conformation, thus preventing SARS-CoV-2 viral entry into cells [101]. Numerous clinical trials investigating supplementation with omega-3 PUFA in COVID-19 are ongoing (ClinicalTrials.gov, accessed on 15 May 2021).

The effect of omega-3 PUFA on SARS-CoV infections could be indirectly associated with the gut microbiota. Studies have shown that supplementation with omega-3 PUFA and its metabolites docosahexaenoic acid improves the composition of the gut microbiota [102,103,104,105]. Indeed, supplementation of omega-3 to healthy subjects increased beneficial *Bifidobacterium*, *Roseburia*, and *Lactobacillus* [106]. Another study also showed that dietary intervention of human subjects with 500 mg of omega-3 increased gut beneficial *Coprococcus* spp. and *Bacteroides* spp. and decreased fatty liver associated *Colinsella* spp. with increased SCFAs [35]. In a rat model, Zhu et al. (2020) showed that omega 3 (flaxseed oil) reduced diabetes patients’ blood glucose through decreasing blood levels of LPS and inflammatory markers IL-6, IL-17A, IL-1β, and TNF-α, as well as increasing SCFAs [107].

## 7. Trace Elements

Trace elements are essential for numerous normal physiological processes, and deficiencies of trace elements can cause various diseases and weaken immune network defenses against pathogenic insults. Several trace element deficiencies have been associated with the severity of COVID-19 [108]. Thus, supplementation with trace elements could reduce the severity of the COVID-19 infection. Alternatively, excessive trace elements could be deleterious to health; therefore, a prudent supplementation approach is important in order to maintain these elements in proportionate and adequate levels.

### 7.1. Zinc

Zinc is a trace element required for many physiological processes such as activation of signaling pathways involved in innate and adaptive immune responses, many enzyme functionalities, and cell membrane permeability. Thus, zinc deficiency (less than 60 mcg/dL) can cause dysfunctions of some physiological processes, leading to disease status [100]. Indeed, many risk factors for severe COVID-19 have been associated with deficiency of zinc such as advanced age, chronic cardiac or pulmonary disease, hypertension, and diabetes [109]. Weakened anti-viral immunity due to the deficiency of zinc could also be a mechanism for these risk factors-causing severe COVID-19 infections. Zinc has been shown to have direct anti-viral effects [109,110].

Studies have shown that zinc is important for the maintenance of gut microbiota homeostasis. In a chick model, chronic zinc depletion reduced species diversity with decreased the levels of SCFAs [111]. With supplementation of zinc to animals challenged with pathogenic bacteria *Salmonella* and *E. Coli*, the number of their pathobionts was reduced with increased abundance of *Lactobacilli* [112,113]. Butyrate concentration was increased with increased IL-10 and decreased IL-6 [113]. However, a study showed that excessive zinc (12-fold of normal mouse chow) caused dysbiosis which increased the severity of *Clostridium difficile* infection [114]. Therefore, adequate dosage of zinc supplementation is critical for its role in gut microbiota homeostasis.

### 7.2. Selenium

Selenium is necessary for normal physiological processes and lack of selenium causes poor immune function [115]. Deficiency of selenium can impair immune cell functionalities and cause inadequate inflammatory responses [116]. Adequate levels of selenium have been associated with good outcomes of COVID-19 while sub-optimal selenium has been found in hospitalized patients [117]. Therefore, supplementation of selenium of patients with COVID-19 to maintain suitable selenium levels could be helpful to avoid severe COVID-19 and reduce the mortality rate [117].

Selenium is closely associated with the homeostasis of gut microbiota and commensal bacterial metabolites. Administration of selenium nanoparticles into chicks increased beneficial bacteria *Lactobacillus* and *Faecalibacterium* with increased production of butyrate [118]. Wei et al. (2019) showed that a hydroxy-analog of selenomethionine was better that sodium selenite, which produced dose-dependent increases of volatile fatty acids, propionate, and butyrate [119]. However, excessive selenium is also a disturbing factor for the gut microbiota. Therefore, it is important to maintain selenium at optimal levels in patients with COVID-19.

### 7.3. Magnesium

Magnesium deficiency has been shown to increase the cytokine storm in COVID-19 [120]. Indeed, magnesium deficiency is common in patients with risk factors of severe COVID-19 such as aging, type 2 diabetes, metabolic syndrome, cardiovascular disease, depression, and hypertension [121,122]. Deficiency of magnesium causes intestinal and systemic inflammation with leukocyte and macrophage activation and increased production of pro-inflammatory cytokines [123,124]. Therefore, supplementation with magnesium may be an important requisite for patients with COVID-19 infections.

Magnesium deficiency as well as oversupply can cause gut dysbiosis. In a mouse model, a magnesium deficient-diet caused altered gut microbiota such as decreased *Bifidobacterum* [124,125,126]. Magnesium deficiency in mice caused increases in both intestinal and systemic inflammatory tone, indicated by increased blood levels of TNF-α and IL6 [126]. In addition, a study showed that oversupply with high dose magnesium to rats which were not deficient in magnesium progressed intestinal dysbiosis [127].

## 8. Conclusions

As intestinal dysbiosis is a common occurrence in the pathogenesis of COVID-19, it is essential that improvement in the health of the gut microbiota is addressed. Various approaches could be helpful in modulating the gut microbiota. Probiotics, prebiotics, and synbiotics could directly and beneficially alter the gut microbiota. However, commensal bacterial metabolites may be more effective in those cases with severe intestinal SARS-CoV-2 infections and/or with the extensive use of antibiotics and other therapeutic agents such as proton pump inhibitors, antidepressants, laxatives, and statins. Supplementation of nutraceuticals and trace elements could facilitate the improvement of the gut microbiota, particularly in those patients proven to have trace element deficits. However, particular attention should be paid to the dosage of trace elements that are administered, given that excessive trace elements can trigger gut dysbiosis. A healthy, diverse, and abundant gut microbiota not only reduces the severity of COVID-19 but also increases the efficacy of vaccines against COVID-19.

## Figures and Tables

**Figure 1 jcm-10-02903-f001:**
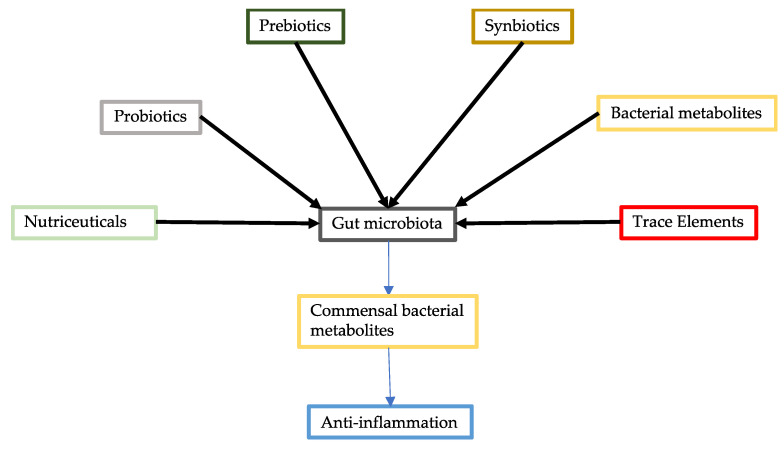
Modulation of gut microbiota. Gut microbiota can be improved through various approaches including administration of probiotics, prebiotics, synbiotics, commensal bacterial metabolites, nutraceuticals, and trace elements. Improved gut microbiota can increase production of commensal bacterial metabolites, which have an anti-inflammatory effect.

**Figure 2 jcm-10-02903-f002:**
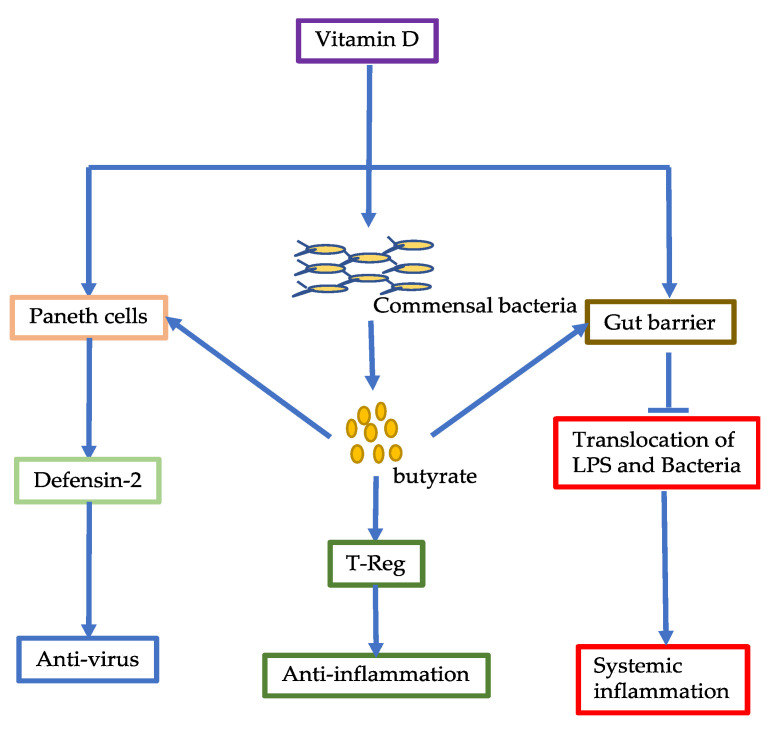
Effects of vitamin D on the intestinal cells, gut barrier, and gut microbiota. Vitamin D can improve gut microbiota, leading to increased production of butyrate. Butyrate can exert anti-inflammatory effects, increase gut barrier, and promote Paneth cells to secret defensin-2. Vitamin D can also directly act on gut barrier to decrease translocation of LPS and bacteria to prevent systemic inflammation. Vitamin D can also promote defensin-2 secretion by Paneth cells to increase anti-viral effect.

**Table 1 jcm-10-02903-t001:** Clinical trials of probiotics for COVID-19 (ClinicalTrials.gov, accessed on 15 May 2021).

Trial Number	Title	Interventions	Country
NCT04621071	Efficacy of probiotics in reducing duration and symptoms of COVID-19 (PROVID-19)	Two probiotic strains (2 strains 10 × 10^9^ cfu) × 25 days Details of probiotics not provided	Canada
NCT04877704	Symprove (Probiotic) as an add-on to COVID-19 management	Probiotic (Symprove) daily for 3 months Details of the probiotic not provided	United Kingdom
NCT04390477	Study to evaluate the effect of a probiotic in COVID-19	Oral daily capsule containing probiotic strains at 1 × 10^9^ cfu for 30 days Details of the probiotic not provided	Spain
NCT04458519	Efficacy of intranasal probiotic treatment to reduce severity of symptoms in COVID19 infection	Nasal irrigations with probiorinse of *Lactococcus Lactis W136* at 2 × 10^9^ cfu twice a day for 14 days	Canada
NCT04366180	Evaluation of the probiotic Lactobacillus coryniformis K8 on COVID-19 prevention in healthcare workers	Dietary one capsule with *Lactobacillus K8* 3 × 10^9^ cfu daily for 2 months	Spain
NCT04734886	The effect of probiotic supplementation on SARS-CoV-2 antibody response after COVID-19	Dietary *L. reuteri DSM 17,938* at 2 × 10^8^ cfu + 60 μg vitamin D daily	Sweden
NCT04666116	Changes in viral load in COVID-19 after probiotics	Dietary supplementation of *Bifidobacterium longum* *Bifidobacterium animalis subsp. lactis* and *Lactobacillus rhamnosus* + vitamin D, zinc, and selenium Dosage not provided	Spain
NCT04847349	Live microbials to boost anti-severe acute respiratory syndrome coronavirus-2 (SARS-CoV-2) immunity clinical trial	Standard or high dose probiotic consortium OL-1 daily for 21 days Details of the probiotic not provided	USA
NCT04756466	Effect of the consumption of a Lactobacillus strain on the incidence of COVID-19 in the elderly	*Lactobacillus* 3 × 10^9^ cfu daily for 3 months	Spain
NCT04366089	Oxygen-ozone as adjuvant treatment in early control of COVID-19 progression and modulation of the gut microbial flora	Azithromycin hydroxychloroquine plus probiotic SivoMixx containing *Streptococcus thermophilus DSM322245 Bifidobacterium lactis DSM 32,246 Bifidobacterium lactis DSM 32,247 Lactobacillus acidophilus DSM 32,241 Lactobacillus helveticus DSM 32,242 Lactobacillus paracasei DSM 32,243 Lactobacillus plantarum DSM 32,244 Lactobacillus brevis DSM 27961* Total mixed probiotics 100 × 10^9^ cfu b.i.d. *	Italy
NCT04517422	Efficacy of L. Plantarum and P. Acidilactici in adults with SARS-CoV-2 and COVID-19	One capsule daily for 30 days (*Lactobacillus plantarum CECT 30,292 Lactobacillus plantarum CECT 7484 Lactobacillus plantarum CECT 7485* *P. acidilactici CECT 7483*) Dosage not provided	Mexico
NCT04462627	Reduction of COVID-19 transmission to health care professionals	Probactiol plus (Metagenics) Containing *L. acidophilus* NCFM at 12.5 × 10^9^ cfu *B. Lactis* Bi-07 at 12.5 × 10^9^ cfu and vitamin D 2.5 μg daily for 3 weeks	Belgium
NCT04420676	Synbiotic therapy of gastrointestinal symptoms during COVID-19 infection	Twice daily of Omnibiotic 10 AAD Containing bacteria (*Bifidobacterium bifidum W23* *Bifidobacterium lactis W51* *Enterococcus faecium W54* *Lactobacillus acidophilus W37* *Lactobacillus acidophilus W55* *Lactobacillus paracasei W20* *Lactobacillus plantarum W1* *Lactobacillus plantarum W62* *Lactobacillus rhamnosus W71* and *Lactobacillus salivarius W24*) and matrix (maize starch, maltodextrin, inulin, potassium chloride, hydrolyzed rice protein, magnesium sulphate, fructooligosaccharide (FOS), enzymes (amylases), vanilla flavor, and manganese sulphate) b.i.d. 11 × 10^9^ cfu for 30 days.	Austria
NCT04399252	Effect of Lactobacillus on the microbiome of household contacts exposed to COVID-19	*Lactobacillus rhamnosus* GG 2 capsules daily for 28 days Bacterial amount in one capsule not provided	USA
NCT04798677	Efficacy and tolerability of ABBC1 in volunteers receiving the influenza or COVID-19 vaccine	ABBC1 immunoessential containing beta 1,3-beta1,6-glucan and a consortium of *C. cerevisae* + selenium + zinc Dosage not provided	Spain
NCT04793997	Microbiome therapy in COVID-19 primary care support	Microbiome spray with 3 beneficial lactobacillus strains for 2 weeks Dosage not provided	Belgium
NCT04507867	Effect of Nss to reduce complications in patients with COVID-19 and comorbidities	Nutrition support system containing *S. boulardii*, prebiotics and omega 3 + zinc + selenium + magnesium + vitamin D_3_ + glutamine Dosage not provided	Mexico
NCT04847349	Live microbials to boost anti-severe respiratory syndrome coronavirus-2 (SARS-CoV-2) immunity clinical trial	OL-1 standard dose or OL-1 high dose. The details of probiotics and dosage not provided	USA
NCT04813718	Post COVID-19 syndrome and the gut-lung axis	Omni-Biotic Pro Vi 5	Austria

* b.i.d. = twice per day.

## Data Availability

www.clinicaltrials.gov, accessed on 15 May 2021.

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
