# Peer review of "Modulation of Gut Microbiota for the Prevention and Treatment of COVID-19"

_jcm, 2021, doi:10.3390/jcm10132903_

Round 1

Reviewer 1 Report

I read with interest this review summarizing evidences on the prevention and treatment of COVID-19 with probiotics. However, before a possible acceptance, the authors should address the following items:

  • It is not clear (neither from the title nor from the introduction) whether it is a systematic or narrative review
  • Introduction: the authors should better explain the relationship between dysbiosis and COVID-19 infection, symptoms and severity (see  10.1111/nmo.14104
  • Probiotics line 58: mechanisms underlying COVID-19 dysbiosis resemble those of many other intestinal inflammation (i.e. quote  10.3390/nu12092674)
  •  Table 1 shows trials concerning only probiotics and synbiotics (no tables concerning prebiotics, trace elements and bacterial metabolites are shown)
    - In table 1 the information regarding the clinical trials cited is scarce (e.g. the numbers of the population willing to be enrolled not reported; the control groups are not reported; type of trial)
    - the severity of the disease is never specified when we talk about the effect of the various modulators of the microbiota (we only talk about severity in general)
    - it is not specified which aspects of the disease have the greatest benefit (in the clinical course) with the addition of microbiota modulators (onset of symptoms?  laboratory? Hospitalization time? Recovery?)
    - there is no mention of the fact that the patient populations examined take other drugs that alter the microbiota and reduce the effect of modulators
  • Nutraceuticals: authors should amend the evaluation of nutritional status of COVID-19 patients (i.e. 10.1007/s40520-020-01727-5) and its relationships with disease severity and dysbiosis and possible therapeutic strategies.

Author Response

  • It is not clear (neither from the title nor from the introduction) whether it is a systematic or narrative review
  • We have added and stated in the abstract and the introduction that the manuscript is a narrative review.
  • Introduction: the authors should better explain the relationship between dysbiosis and COVID-19 infection, symptoms and severity (see 10.1111/nmo.14104
  • We have added the suggested reference and noted dysbiosis in relation to COVID-19.
  • Probiotics line 58: mechanisms underlying COVID-19 dysbiosis resemble those of many other intestinal inflammation (i.e., quote 10.3390/nu12092674)
  • We have added the suggested reference and noted that dysbiosis in COVID-19 resembles those of other intestinal diseases.
  •  Table 1 shows trials concerning only probiotics and synbiotics (no tables concerning prebiotics, trace elements and bacterial metabolites are shown)
    - In table 1 the information regarding the clinical trials cited is scarce (e.g. the numbers of the population willing to be enrolled not reported; the control groups are not reported; type of trial)
    - the severity of the disease is never specified when we talk about the effect of the various modulators of the microbiota (we only talk about severity in general)
    - it is not specified which aspects of the disease have the greatest benefit (in the clinical course) with the addition of microbiota modulators (onset of symptoms?  laboratory? Hospitalization time? Recovery?)
    - there is no mention of the fact that the patient populations examined take other drugs that alter the microbiota and reduce the effect of modulators
  • We have noted those clinical trials that were uploaded to ClinicalTrials.gov in order to site the number of trials that show interest in investigating probiotics and synbiotics in COVID-19 infections and not other factors. We focus on what sort of probiotic bacteria are used in these clinical trials which facilitate our discussion about how a probiotic improves gut microbiota and thus decreases hyperinflammation, a characteristic of severe COVID-19.
  • In our narrative review, we define the severity of the disease by pathophysiological process, i.e. hyperinflammation. Gut dysbiosis promotes hyperinflammation through increased proinflammatory factors and reduced anti-inflammatory mechanisms.  Therefore, modulation of the gut microbiota could reduce proinflammatory status and increase anti-inflammatory mechanisms through increased beneficial bacteria and commensal bacterial metabolites.
  • We mainly refer to decreased hyperinflammation by modulators, which leads to decreased severity of the disease including relieved symptoms,  improved lab indicators, shortened hospitalization time and better recovery from the disease.
  • We have also included other therapeutic agents that alter gut microbiota such as proton pump inhibitors, antidepressants, laxatives and statin (PMID: 28118083, PMID: 31953381).
  • Nutraceuticals: authors should amend the evaluation of nutritional status of COVID-19 patients (i.e., 10.1007/s40520-020-01727-5) and its relationships with disease severity and dysbiosis and possible therapeutic strategies.
  • We have added the suggested reference with an additional sentence. See line 245.

Reviewer 2 Report

This MS discusses aspects of the intestinal microbiome in relevance to COVID-19 

SPECIFIC COMMENTS

  1. There are various errors of language that could be enhanced. For two examples: the first sentence of the ABSTRACT and line 273/274
  2. The term "COVID-19 patients" needs to be corrected to read "patients with COVID-19"
  3. Bacterial terms need to be correctly displayed, following standard convention (also error on page 2)
  4. NCT04420676 in Table 2 should say Austria
  5. At line 219, one assumes that this should say probiotics and prebiotics individually
  6. Line 270 should read: "Gavioli et al. [77] also showed...." Please check that all references are placed correctly

Author Response

  • There are various errors of language that could be enhanced. For two examples: the first sentence of the ABSTRACT and line 273/274
  • We have amended and rectified the sentences and also have carefully read through the manuscript.
  • The term "COVID-19 patients" needs to be corrected to read "patients with COVID-19"
  • We have amended the term as suggested.
  • Bacterial terms need to be correctly displayed, following standard convention (also error on page 2)
  • We have corrected bacterial terms especially on page 2.
  • NCT04420676 in Table 2 should say Austria
  • We have amended Table 1.
  • At line 219, one assumes that this should say probiotics and prebiotics individually
  • We have amended as advised.
  • Line 270 should read: "Gavioli et al. [77] also showed...." Please check that all references are placed correctly
  • We have checked and amended as advised.

Round 2

Reviewer 1 Report

In the revised version the authors amended all issues raised before and improved the quality of the manuscript